# Sharing neurophysiology data from the Allen Brain Observatory

**Saskia EJ de Vries\*†, Joshua H Siegle\*†, Christof Koch**

Allen Institute, Seattle, United States

**Abstract** Nullius in verba ('trust no one'), chosen as the motto of the Royal Society in 1660, implies that independently verifiable observations—rather than authoritative claims—are a defining feature of empirical science. As the complexity of modern scientific instrumentation has made exact replications prohibitive, sharing data is now essential for ensuring the trustworthiness of one's findings. While embraced in spirit by many, in practice open data sharing remains the exception in contemporary systems neuroscience. Here, we take stock of the Allen Brain Observatory, an effort to share data and metadata associated with surveys of neuronal activity in the visual system of laboratory mice. Data from these surveys have been used to produce new discoveries, to validate computational algorithms, and as a benchmark for comparison with other data, resulting in over 100 publications and preprints to date. We distill some of the lessons learned about open surveys and data reuse, including remaining barriers to data sharing and what might be done to address these.

## Editor's evaluation

This article presents an important review of data-sharing efforts in neurophysiology, with a focus on data released by the Allen Institute for Brain Science. The article offers perspectives from the users of such shared data, and makes a compelling case that data sharing has already advanced research in neuroscience. There are valuable insights here for producers and users of neurophysiology data, as well as the funders that support all those efforts.

**\*For correspondence:**
saskiad@alleninstitute.org
(SEJdV);
joshs@alleninstitute.org (JHS)

†These authors contributed equally to this work

**Competing interest:** The authors declare that no competing interests exist.

## Introduction

Why share data? The central nervous system is among the most complex organs under investigation. Accordingly, the tools to study it have become intricate and costly, generating ever-growing torrents of data that need to be ingested, quality-controlled, and curated for subsequent analysis. Not every lab has the financial or personnel resources to accomplish this. Moreover, while many scientists relish running experiments, others find their passion in analysis. Data collection requires a different skillset than analysis, especially as the field demands more comprehensive and higher-dimensional datasets, which, in turn, necessitate more advanced analytical methods and software infrastructure. A scientific ecosystem in which data is extensively shared and reused would give researchers more freedom to focus on their favorite parts of the discovery process.

Sharing data brings other benefits as well. It increases the number of eyes on each dataset, making it easier to spot potential outlier effects (*Button et al., 2013*). It encourages meta-analyses that integrate data from multiple studies, providing the opportunity to reconcile apparently contradicting results or expose the biases inherent in specific analysis pipelines (*Botvinik-Nezer et al., 2020*; *Mesa et al., 2021*). It also gives researchers a chance to test hypotheses on existing data, refining and updating their ideas before embarking on the more costly process of running new experiments.

Without a doubt, reanalysis of neurophysiology data has already facilitated numerous advances. Electrophysiological recordings from nonhuman primates, which require tremendous dedication to collect, are often reused in multiple high-impact publications (*Churchland et al., 2010*; *Murray*

*et al., 2014*). Data from 'calibration' experiments, in which activity of individual neurons is monitored via two modalities at once, have been extremely valuable for improving data processing algorithms (*GENIE Project, 2015*; *Henze et al., 2009*; *Huang et al., 2021*; *Neto et al., 2016*). A number of these datasets have been shared via the website of CRCNS (*Teeters et al., 2008*), far-sighted organization focused on aggregating data for computational neuroscience within the same searchable database. To date, CRCNS hosts 150 datasets, including extensive neurophysiology recordings from a variety of species, as well as fMRI, EEG, and eye movement datasets. This is especially impressive given that CRCNS was launched by a single lab in 2008. The repository does not enforce formatting standards, and thus each dataset differs in its packaging conventions, as well as what level of preprocessing may have been applied to the data. The website includes a list of 111 publications and preprints based on CRCNS data. Our own meta-analysis of these articles shows that 28 out of 150 datasets have been reused at least once, with four reused more than 10 times each.

More recently, an increasing number of researchers are choosing to make data public via generalist repositories such as Figshare, Dryad, and Zenodo, or the neuroscience-specific G-Node Infrastructure. In addition, the lab of György Buzsáki maintains a databank of recordings from more than 1000 sessions from freely moving rodents (*Petersen et al., 2020*). As data can be hosted on these repositories for free, they greatly lower the barriers to sharing. However, the same features that reduce the barriers for sharing can also increase the barriers for reuse. With no restrictions on the data format or level of documentation, learning how to analyze diverse open datasets can take substantial effort, and scientists are limited in their ability to perform meta-analyses across datasets. Further, with limited and nonstandard documentation, finding relevant datasets can be challenging.

Since its founding, the Allen Institute has made open data one of its core principles. Specifically, it has become known for generating and sharing *survey* datasets within the field of neuroscience, taking inspiration from domains such as astronomy where such surveys are common. (As a community, astronomers have developed a far more comprehensive and coherent data infrastructure than biology. One obvious reason is the existence of a single sky with an agreed-upon coordinate system and associated standards such as the Flexible Image Transport System; *Borgman et al., 2016*; *York et al., 2000*; *Zuiderwijk and Spiers, 2019*.) The original Allen Mouse Brain Atlas (*Lein et al., 2007*) and subsequent surveys of gene expression (*Bakken et al., 2016*; *Hawrylycz et al., 2012*; *Miller et al., 2014*), mesoscale connectivity (*Harris et al., 2019*; *Oh et al., 2014*), and in vitro firing patterns (*Gouwens et al., 2019*) have become essential resources across the field. These survey datasets are (1) collected in a highly standardized manner with stringent quality controls, (2) create a volume of data that is much larger than typical individual studies within their particular disciplines, and (3) are collected without a specific hypothesis to facilitate a diverse range of use cases.

Starting a decade ago, we began planning the first surveys of in vivo physiology in mouse cortex with single-cell resolution (*Koch and Reid, 2012*). Whereas gene expression and connectivity are expected to change relatively slowly, neural responses in awake subjects can vary dramatically from moment to moment, even during apparently quiescent periods (*McCormick et al., 2020*). Therefore, an in vivo survey of neural activity poses new challenges, requiring many trials and sessions to account for both intra- as well as inter-subject variability. We first used two-photon calcium imaging and later Neuropixels electrophysiology to record spontaneous and evoked activity in visual cortex and thalamus of awake mice that were passively exposed to a wide range of visual stimuli (known as 'Visual Coding' experiments). A large number of subjects, highly standardized procedures, and rigorous quality control criteria distinguished these surveys from typical small-scale neurophysiology studies. More recently, the Institute carried out surveys of single-cell activity in mice performing a visually guided behavioral task (known as 'Visual Behavior' experiments). In all cases, the data was shared even before we published our own analyses of them. We reflect here on the lessons learned concerning the challenges of data sharing and reuse in the neurophysiology space. Our primary takeaway is that the widespread mining of our publicly available resources demonstrates a clear community demand for open neurophysiology data and points to a future in which data reuse becomes more commonplace. However, more work is needed to make data sharing and reuse practical (and ideally the default) for all laboratories practicing systems neuroscience.

## Overview of the Allen Brain Observatory

The Allen Brain Observatory consists of a set of standardized instruments and protocols designed to carry out surveys of cellular-scale neurophysiology in awake brains (*de Vries et al., 2020*; *Siegle et al., 2021a*). Our initial focus was on neuronal activity in the mouse visual cortex (*Koch and Reid, 2012*). Vision is the most widely studied sensory modality in mammals, but much of the foundational work is based on recordings with hand-tuned stimuli optimized for individual neurons, typically investigating a single area at a time (*Hubel and Wiesel, 1998*). The field has lacked the sort of unbiased, large-scale surveys required to rigorously test theoretical models of visual function (*Olshausen and Field, 2005*). The laboratory mouse is an advantageous model animal given the extensive ongoing work on mouse cell types (*BRAIN Initiative Cell Census Network (BICCN), 2021*; *Tasic et al., 2018*; *Yao et al., 2021*; *Zeisel et al., 2015*), as well as access to a well-established suite of genetic tools for observing and manipulating neural activity via driver and reporter lines or viruses (*Gerfen et al., 2013*; *Madisen et al., 2015*).

Our two-photon calcium imaging dataset (*Allen Institute Mindscope Program, 2016*) leveraged transgenic lines to drive the expression of a genetically encoded calcium indicator (*Chen et al., 2013*) in specific populations of excitatory neurons (often constrained to a specific cortical layer) or GABAergic interneurons. In total, we recorded activity from over 63,000 neurons across 6 cortical areas, 4 cortical layers, and 14 transgenic lines (*Figure 1*). The Neuropixels electrophysiology dataset (*Allen Institute MindScope Program, 2019*) used silicon probes (*Jun et al., 2017*) to record simultaneously from the same six cortical areas targeted in the two-photon dataset, as well as additional subcortical regions (*Durand et al., 2022*). While cell type specificity was largely lost, transgenic lines did enable optotagging of specific inhibitory interneurons. The Neuropixels dataset included recordings from over 40,000 units passing quality control across more than 14 brain regions and 4 mouse lines (*Figure 1*). In both surveys, mice were passively exposed to a range of visual stimuli. These included drifting and flashed sinusoidal gratings to measure traditional spatial and temporal tuning properties, sparse noise or windowed gratings to map spatial receptive fields, images and movies that have natural spatial and temporal statistics, and epochs of mean luminance to capture neurons' spontaneous activity. These stimuli were selected to provide a broad survey of visual physiological activity and compare the organization of visual responses across brain regions and cell types. Mice were awake during these experiments and head-fixed on a spinning disk that permitted them to run in a self-initiated and unguided manner. Subsequent surveys of neural activity in mice performing a behavioral task are not discussed here as it is too soon to begin evaluating their impact on the field.

## Approach to data distribution

Once the data was collected, we wanted to minimize the friction required for external groups to access it and mine it for insights. This is challenging! Providing unfettered access to the data can be accomplished by providing a simple download link; yet, unless the user understands what is contained in the file and has installed the appropriate libraries for parsing the data, its usefulness is limited. At the other extreme, a web-based analysis interface that does not require any downloading or installation can facilitate easy data exploration, but this approach has high upfront development costs and imposes limitations on the analyses that can be carried out.

These conflicting demands are apparent in our custom tool, the AllenSDK, a Python package that serves as the primary interface for downloading data from these surveys as well as other Allen Institute resources. In the case of the Allen Brain Observatory, the AllenSDK provides wrapper functions for interacting with the Neurodata Without Borders (NWB) files (*Rübel et al., 2022*; *Teeters et al., 2015*) in which the data is stored. Intuitive functions enable users to search metadata for specific experimental sessions and extract the relevant data assets. Whereas our two-photon calcium imaging survey was accompanied by a dedicated web interface that displayed summary plots for every cell and experiment (observatory.brain-map.org/visualcoding), we discontinued this practice because of its associated development costs and because most users preferred to directly access the data in their own analysis environment.

One challenge with sharing cellular neurophysiology data is that it includes multiple high-dimensional data streams. Many other data modalities (e.g., gene expression) can be reduced to a derived metric and easily shared in a tabular format (e.g., *cell-by-gene* table). In contrast, neurophysiological data

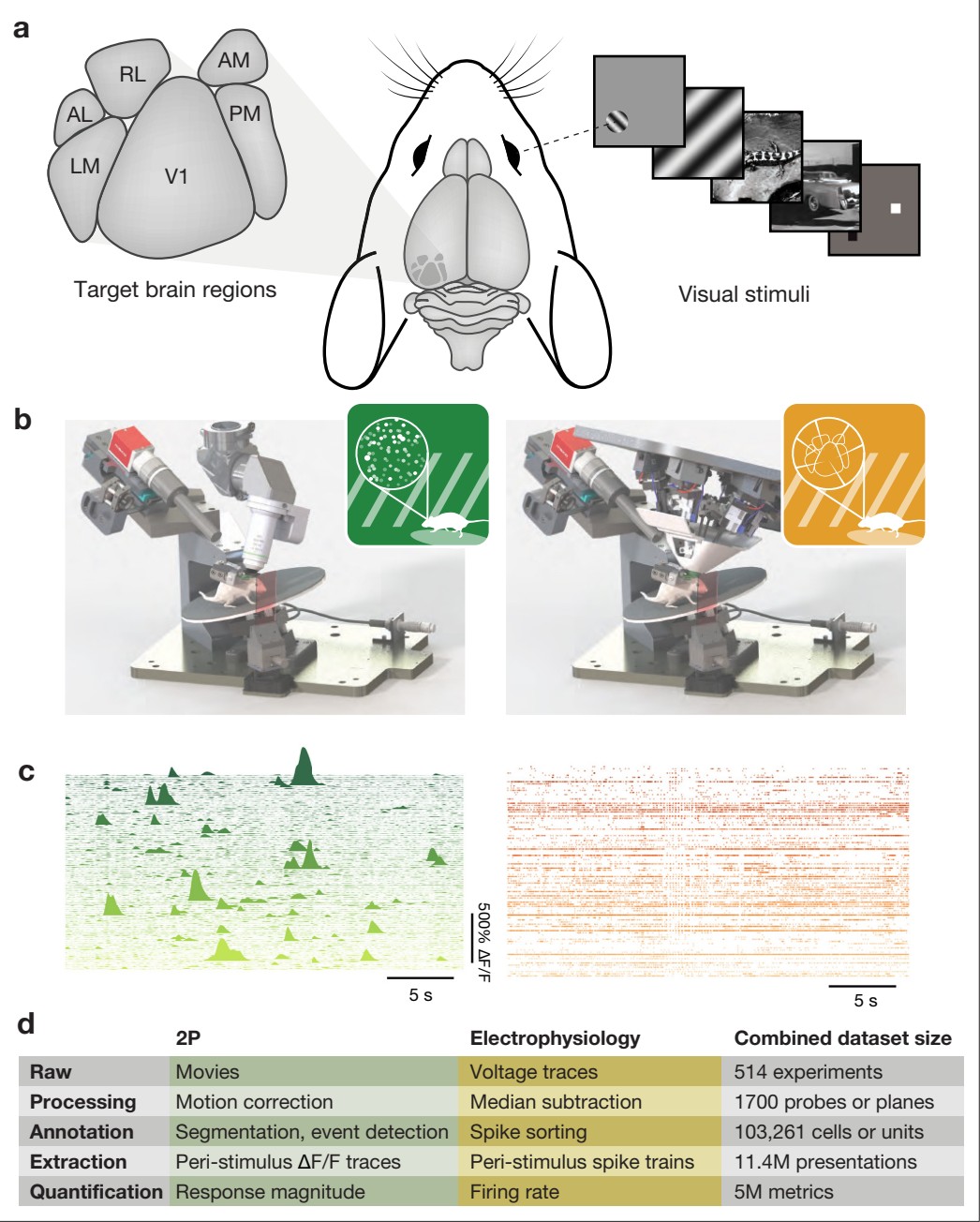

**Figure 1.** Overview of Allen Brain Observatory Visual Coding datasets. (**a**) Target brain regions and example visual stimuli. (**b**) Standardized rigs for two-photon calcium imaging (left) and Neuropixels electrophysiology (right). (**c**) Example ΔF/F traces or spike rasters for 100 simultaneously recorded neurons from each modality. Both are extracted during one presentation of a 30 s natural movie. (**d**) Dataset size after different stages of analysis.

The online version of this article includes the following source data for figure 1:

**Source data 1.** Dataset size after each stage of analysis.

is highly varied, with researchers taking different approaches to both data processing (e.g., spike sorting or cell segmentation) and analysis. While these data can be analyzed as a large collection of single-cell recordings, they can also be approached as population recording, leveraging the fact that hundreds to thousands of neurons are recorded simultaneously. Thus, particularly for a survey-style dataset not designed to test a particular hypothesis, it is hard to reduce these recordings to a simple set of derived metrics that encapsulate the full range of neural and behavioral states. Even when it is

possible (e.g., we could have shared a table of single-cell receptive field and tuning properties as the end product), this confines any downstream analyses to those specific metrics, severely undermining the space of possible use cases. At the same time, if we had only shared the raw data, few researchers would have the resources or the inclination to build their own preprocessing and packaging pipelines.

Therefore, we aimed to share our data in a flexible way to facilitate diverse use cases. For every session, we provided either spike times or fluorescence traces, temporally aligned stimulus information, the mouse's running speed and pupil tracking data, as well as intermediate, derived data constructs, such as ROI masks, neuropil traces, and pre- and post- de-mixing traces for two-photon microscopy, and waveforms across channels for Neuropixels. All are contained within the NWB files. In addition, we uploaded the more cumbersome, terabyte-scale raw imaging movies and voltage traces to the public cloud for users focused on data processing algorithms (*Figure 2*).

## Three families of use cases

The first round of two-photon calcium imaging data was released in July 2016, followed by three subsequent releases that expanded the dataset (green triangles in *Figure 3*). The Neuropixels dataset became available in October 2019 (yellow triangle in *Figure 3*). At the end of 2022, there were 104 publications or preprints that reuse these two datasets, with first authors at 50 unique institutions. This demonstrates the broad appeal of applying a survey-style approach to the domain of in vivo neurophysiology.

We found three general use cases of Allen Brain Observatory data in the research community:

1. Generating novel discoveries about brain function
2. Validating new computational models and algorithms
3. Comparing with experiments performed outside the Allen Institute

Below, we highlight some examples of these three use cases, for both the two-photon calcium imaging and Neuropixels datasets. All these studies were carried out by groups external to the Allen Institute, and frequently without any interaction from us, speaking to the ease with which data can be downloaded and analyzed.

### Making discoveries

*Sweeney and Clopath, 2020* used Allen Brain Observatory two-photon imaging data to explore the stability of neural responses over time. They previously found that neurons in a recurrent network model with high inherent plasticity had more variability in their stimulus selectivity than those with low plasticity. They also found that neurons with high inherent plasticity have higher population coupling. To examine whether these were related, they here analyzed real calcium-dependent fluorescence traces from the Allen Brain Observatory to examine whether population coupling and response variability were correlated. The authors found that, indeed, population coupling is correlated with the change in orientation and direction tuning of neurons over the course of a single experiment, an unexpected result linking population activity with individual neural responses.

*Bakhtiari et al., 2021* examined whether a deep artificial neural network (ANN) could model both the ventral and dorsal pathways of the visual system in a single network with a single cost function. They trained two networks, one with a single pathway and the other with two parallel pathways, using a Contrastive Predictive Coding loss function. Comparing the representations of these networks with the neural responses in the two-photon imaging dataset, they found that the single pathway produced ventral-like representations but failed to capture the representational similarity of the dorsal areas. The parallel pathway network, though, induced distinct representations that mapped onto the ventral/dorsal division. This work is an illustration of how large-scale data can guide the development of neural network modeling, and, conversely, how those approaches can inform our understanding of cortical function.

*Fritsche et al., 2022* analyzed the time course of stimulus-specific adaptation in 2365 neurons in the Neuropixels dataset and discovered that a single presentation of a drifting or static grating in a specific orientation leads to a reduction in the response to the same visual stimulus up to eight trials (22 s) in the future. This stimulus-specific, long-term adaptation persists despite intervening stimuli, and is seen in all six visual cortical areas, but not in visual thalamic areas (LGN and LP), which returned

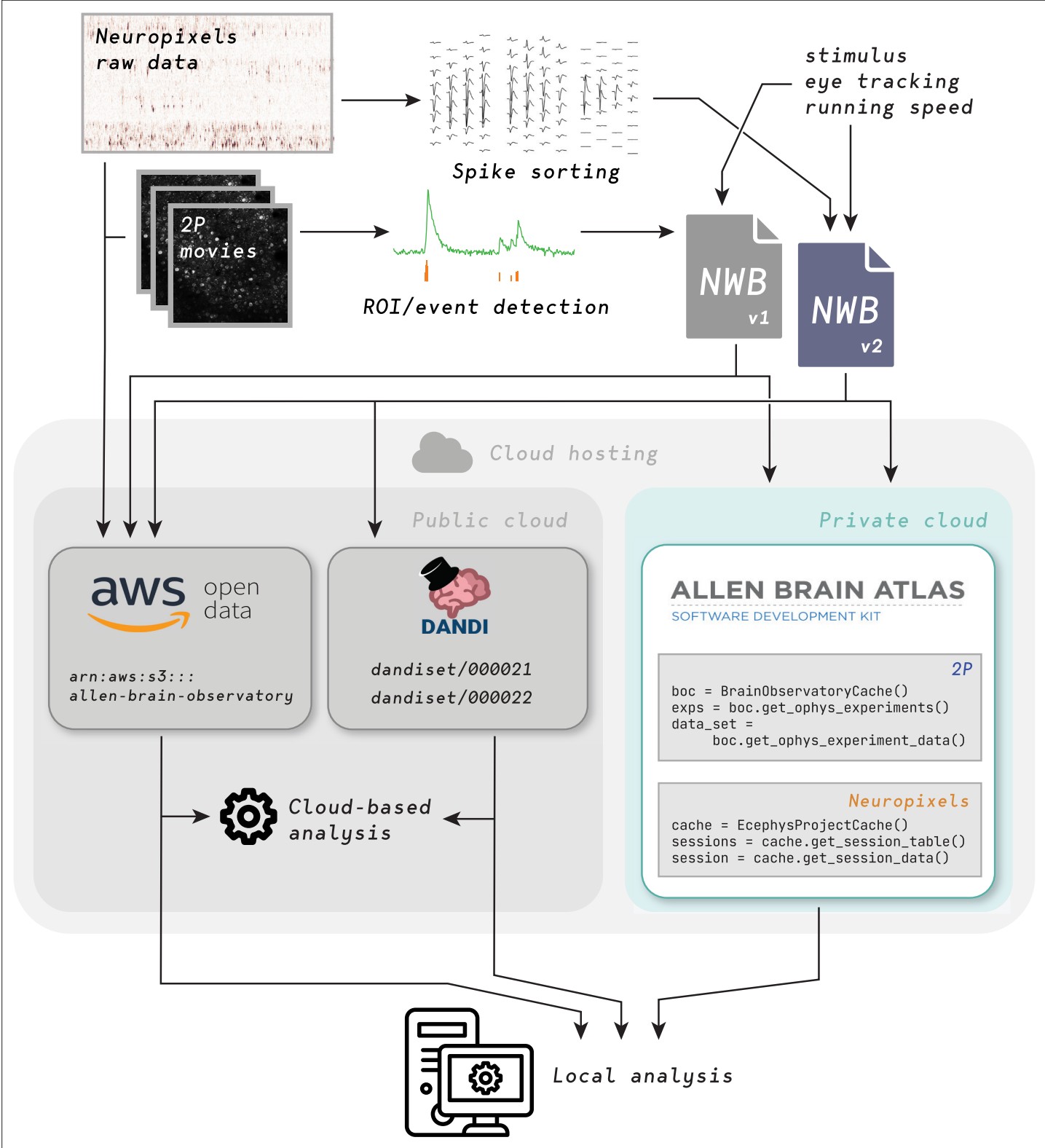

**Figure 2.** Distributing data from Allen Brain Observatory Visual Coding experiments. Raw data is acquired and processed at the Allen Institute, combined with metadata (including 3-D neuronal coordinates, stimulus information, eye-tracking data, and running speed) and packaged into NWB files. Each such file is intended to be a complete, self-contained data record for one experimental session in one animal. NWB files are uploaded to three different locations in the cloud: The Amazon Web Services (AWS) Registry of Open Data, the Distributed Archives for Neurophysiology Data Integration (DANDI) repository, and the Allen Institute data warehouse (accessed via the AllenSDK, a Python API for searching for relevant sessions and

*Figure 2 continued on next page*

*Figure 2 continued*

downloading data). Raw data is also uploaded to the AWS Registry of Open Data. End users can either analyze data in the cloud or download data for local analysis.

---

to baseline after one or two trials. This is a remarkable example of a discovery that was not envisioned when designing our survey, but for which our stimulus set was well suited.

At least three publications have taken advantage of the fact that every Neuropixels insertion targeting visual cortex and thalamus also passed through the intervening hippocampus and subiculum. *Nitzan et al., 2022* analyzed the local field potential from these electrodes to detect the onset of sharp-wave ripples, fast oscillations believed to mediate offline information transfer out of the hippocampus (*Girardeau and Zugaro, 2011*). They found that sharp-wave ripples coincided with a transient, cortex-wide increase in functional connectivity with the hippocampus. *Jeong et al., 2023* examined the topography of this functional connectivity and found that distinct but intermingled classes of visual cortex neurons were preferentially modulated by ripples originating in dorsal hippocampus, while others were more coupled to ripples in intermediate hippocampus. *Purandare and Mehta, 2023* analyzed the responses of hippocampal neurons to natural movies and found that many

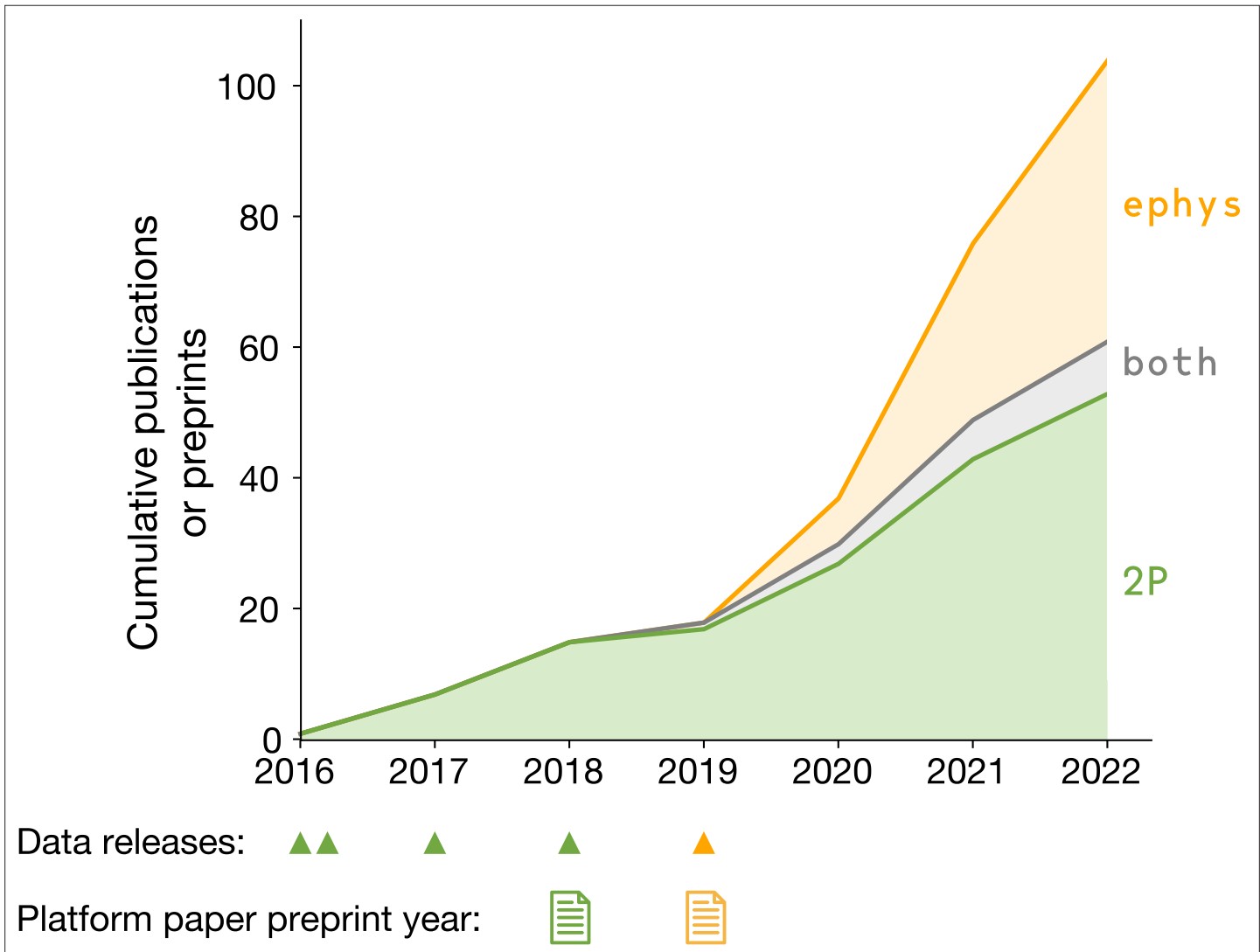

**Figure 3.** Data reuse over time. Cumulative number of papers or preprints that include novel analysis of Allen Brain Observatory Visual Coding surveys. Triangles indicate the years in which new data was made publicly available. Paper icons indicate the years in which the Allen Institute preprints describing the dataset contents and initial scientific findings were posted.

displayed highly selective 'movie fields' that were often as robust as those of neurons in visual cortex. However, in contrast to visual cortex, the movie fields in the hippocampus disappeared if the movie frames were shuffled (thereby disrupting the learned temporal sequence). Although the Allen Brain Observatory experiments were not originally designed to test hypotheses of hippocampal function, the Neuropixels dataset turned out to be attractive for understanding the interactions between this structure and visual cortical and thalamic regions.

## Validating models and algorithms

Many researchers used the numerous and diverse fluorescence movies in the two-photon imaging dataset to validate image processing algorithms. As the different transgenic lines used in the dataset target different populations of neurons, they have different labeling densities. As a result, there are some very sparse movies with only a dozen neurons within the field of view and others with up to ~400 neurons. This makes the dataset a rich resource for benchmarking methods for cell segmentation (*Bao et al., 2021*; *Inan et al., 2021*; *Kirschbaum et al., 2020*; *Petersen et al., 2018*; *Soltanian-Zadeh et al., 2019*), matching neurons across multiple sessions (*Sheintuch et al., 2017*), and removing false transients in the fluorescence traces (*Bao et al., 2022*; *Gauthier et al., 2022*).

*Montijn et al., 2021* used the Neuropixels survey to showcase a novel method for identifying statistically significant changes in neural activity. Their method, called *ZETA* (Zenith of Event-based Time-locked Anomalies), detects whether a cell is responsive to stimulation without the need to tune parameters, such as spike bin size. As an example, they analyze the 'optotagging' portion of the Neuropixels experiments carried out in Vip-Cre × ChR2 mice, involving the activation of Vip+ interneurons with brief pulses of blue light. Intended to aid in the identification of genetically defined cell types at the end of each recording session, the authors show how these recordings can be exploited to test the network-level impact of triggering a particular class of interneurons. ZETA identifies not only Vip+ neurons that are directly activated by the light pulses, but also nearby cortical neurons that are inhibited on short timescales and disinhibited over longer timescales.

*Buccino et al., 2020* used raw data from the Neuropixels survey to validate *SpikeInterface*, a Python package that runs multiple spike sorting algorithms in parallel and compares their outputs. We originally performed spike sorting with one such algorithm, *Kilosort 2* (*Pachitariu et al., 2016*). The authors of this paper used SpikeInterface to compare the performance of Kilosort 2 and five additional algorithms. In one example session, over 1000 distinct units were detected by only one sorter, while only 73 units were detected by five or more sorters. At first glance, this finding seems to indicate a high level of disagreement among the algorithms. However, when comparing these results with those from simulations, it became clear that the low-agreement units were mainly false positives, while the true positive units were highly consistent across algorithms. This finding, and the SpikeInterface package in general, will be essential for improving the accuracy of spike sorting in the future.

## Comparisons with other datasets

*Kumar et al., 2021* used supervised and semi-supervised learning algorithms to classify cortical visual areas based on either spontaneous activity or visually evoked responses. Cortical visual areas, defined based on retinotopic maps, are thought to serve distinct visual processing functions. Rather than compare tuning properties of neurons across the areas, as many studies (including our own) have done, the authors trained classifiers to successfully determine the area membership and boundaries from the neural responses to visual stimuli. They compared the performance of these algorithms for their own wide-field imaging dataset with our two-photon imaging dataset. This provides an extension and validation of their results to conditions in which single-cell responses are available.

*Muzzu and Saleem, 2021* performed electrophysiological recordings in mouse cortex to examine 'mismatch' responses, where neurons respond to differences in visual cue and motor signals from running. The authors argued that these responses derive from visual features rather than the mismatch, showing that these perturbation responses might be explained by preferential tuning to low temporal frequencies. The authors use our two-photon imaging dataset to demonstrate a difference in temporal frequency tuning across cortical layers, with neurons in superficial layers being tuned to lower frequencies, supporting the fact that mismatch responses are predominantly observed in superficial layers. While this use case is perhaps one of the simplest, it is an elegant demonstration of gaining validation for implications that emerge from one's own experiments.

*Stringer et al., 2021* compared spiking activity from the Neuropixels dataset to calcium-dependent fluorescence changes recorded in their laboratory. Their analysis focused on the precision with which the orientation of static gratings can be decoded from activity in visual cortex. Using their own two-photon calcium imaging dataset that consisted of up to 50,000 simultaneously recorded neurons, they found that it was possible to use neural activity to discriminate orientations that differ by less than 0.4°, about a factor of 100 better than reported behavioral thresholds in mice. As an important control, they showed that the trial-to-trial variability in evoked responses to static gratings was nearly identical between their two-photon data and our Neuropixels electrophysiology data, indicating that their main result was not likely to depend on the recording modality. This use case is noteworthy because the preprint containing this comparison appeared less than a month after our dataset became publicly available.

*Schneider et al., 2021* directly compared the Allen Neuropixels dataset with Neuropixels recordings from LGN and V1 carried out locally. They first analyzed gamma-band coherence between these two structures in the Allen Brain Observatory dataset and found evidence in support of their hypothesis that inter-regional coherence is primarily driven by afferent inputs. This contrasts with the 'communication through coherence' hypothesis (*Fries, 2015*), which posits that pre-existing inter-regional coherence is necessary for information transfer. They then performed a separate set of Neuropixels recordings in which they found that silencing cortex (via optogenetic activation of somatostatin-positive interneurons) did not change the degree of coherence between LGN and V1, indicating that V1 phase-locking is inherited from LGN, further supporting their hypothesis. This is an insightful example of how a survey dataset can be used to test a hypothesis, followed by a set of more specific follow-up experiments that refine the initial findings.

## Use in education

These surveys have also been used in a variety of educational contexts. Many computational neuroscience summer courses have presented them as potential source of student projects. This includes the Allen Institute's own *Summer Workshop on the Dynamic Brain* as well as the Cold Spring Harbor *Neural Data Science* and *Computational Neuroscience: Vision* courses; Brains, Minds, and Machines Summer Course at the Marine Biological Laboratory; and the Human Brain Project Education Program. Indeed, in some cases these projects have led to publications (*Christensen and Pillow, 2022*; *Conwell et al., 2022*). Beyond these summer courses, these datasets are discussed in undergraduate classrooms, enabling students to learn computational methods with real data rather than toy models. This includes classes at the University of Washington, Brown University, and the University of California, San Diego.

## User experience

To gain additional insight into the perspectives of end users, we interviewed eight scientists who published papers based on Allen Brain Observatory data. There were three primary reasons why users chose to analyze these datasets: (1) they were interested in the datasets' unique features, such as the number of recorded regions; (2) they lacked the ability to collect data from a particular modality (e.g., an imaging lab wanted to analyze electrophysiology data); or (3) they wanted to validate their own findings using an independent dataset. Although most users initially tried to access the data via the AllenSDK Python package, several found it easier to download the NWB files directly after exporting a list of URLs, particularly if they were using Matlab for analysis. Common challenges included slow data download speeds, understanding the details of preprocessing steps, and data format changes (e.g., the original Neuropixels files were subsequently updated to adhere to the latest NWB standard, which broke compatibility with older versions of the AllenSDK). In most cases, reaching out to scientists at the Allen Institute cleared up these issues. Users also encountered obstacles related to the scale of the data: some scientists needed to learn how to submit jobs to their local high-performing computing cluster to speed-up analysis or to develop new methods for retrieving and organizing data. But the size of the dataset was also one of its biggest advantages:

"From a scientific perspective, facing such a rich dataset can be overwhelming at the beginning—there are so many questions that could be addressed with it and it's easy to get lost. In my case, it was a blessing rather than a curse; my initial question was simply how different areas included in the dataset are modulated by ripples. Having such a wide coverage of the hippocampal axis, I later asked

> ## Box 1. Web resources for Allen Brain Observatory Visual Coding datasets
>
> *White papers describing the surveys*
> 2P – http://help.brain-map.org/display/observatory/Documentation
> Neuropixels – https://portal.brain-map.org/explore/circuits/visual-coding-neuropixels
> *Code repositories*
> AllenSDK – https://github.com/alleninstitute/allensdk
> 2P – https://github.com/AllenInstitute/visual_coding_2p_analysis
> Neuropixels – https://github.com/AllenInstitute/neuropixels_platform_paper
> *Tutorials*
> 2P – https://allensdk.readthedocs.io/en/latest/brain_observatory.html
> Neuropixels – https://allensdk.readthedocs.io/en/latest/visual_coding_neuropixels.html

myself whether ripples recorded on different probes differentially modulate neuronal activity outside of the hippocampus, which led me to some interesting and unexpected findings." (Noam Nitzan, NYU)

In general, researchers were enthusiastic about this resource:

"I looked at several open data sets, and I quickly realized that the Allen Brain Observatory Neuropixels data set was the best documented open data set I found. The intuitive packaging in the NWB format, as well as the systematic repetition of experiments with a comparatively high number of mice and single units in various visual areas, made the decision to use the Allen dataset very easy." (Marius Schneider, Ernst Strüngmann Institute)

Journal referees seemed to respond positively to the use of Allen Brain Observatory data, although one user reported that a reviewer was concerned about their ability to adequately validate data they did not collect themselves. For future data releases, several users requested experiments with different types of visual stimuli, ideally chosen through interactions with the wider community.

## Discussion

Although it is too early to assess the long-term relevance of the first two Allen Brain Observatory datasets, the more than 100 publications that mined this data over the last 6 years testify to its immediate impact. Our data has been used for a wide array of applications, many of which we did not envision when we designed the surveys. We attribute this success to several factors, including the scale of the dataset (tens of thousands of neurons across hundreds of subjects), our extensive curation and documentation efforts (in publications, white papers, and websites), a robust software kit for accessing and analyzing the data (the AllenSDK), and a well-organized outreach program (involving tutorials at conferences and a dedicated summer workshop).

One key lesson we learned is to facilitate different types of data reuse, as illustrated by the examples above. While many users primarily care about spike times or fluorescence traces, others require raw data. Because of this, it was fortuitous that we provided access to both (*Figure 2*). Sharing the data in a way that is flexible and well documented reduces constraints on which questions can be addressed, and is thus paramount for facilitating reuse. Indeed, while many papers leveraged the datasets to examine the visual functional properties of neurons or brain areas, many others used these data in a way that was agnostic to the visual context of the underlying experiments.

We hope to see the sharing of both raw and processed cellular physiology data soon become ubiquitous. However, we know that our surveys were contingent on the efforts of a large team, including scientists from multiple disciplines, hardware and software engineers, research associates, and project managers. Assembling similar resources is untenable for most academic labs. Fortunately, there are ongoing developments that will lower the barriers to sharing and reusing data: *increased standardization* and *cloud-based analysis tools*.

## Increased standardization

The success of data reuse rests on the FAIR Principles: data must be Findable, Accessible, Interoperable, and Reusable (*Wilkinson et al., 2016*). In other words, prospective analysts must be able to easily identify datasets appropriate to their needs and know how to access and use the data assets. This is best accomplished if data is stored in standardized formats, with common conventions for rich metadata and easy-to-use tools for search and visualization.

The Allen Institute has invested heavily in developing and promoting *Neurodata Without Borders* (NWB) as a standard data model and interchange format for neurophysiology data (*Rübel et al., 2022*; *Teeters et al., 2015*). NWB has been criticized for being both too restrictive (as it often takes a dedicated programmer to generate format-compliant files from lab-specific data) and not restrictive enough (as it does not enforce sufficient metadata conventions, especially related to behavioral tasks). Nevertheless, there are overwhelming advantages to having common, language-agnostic formatting conventions across the field. Building a rich ecosystem of analysis and visualization tools based on NWB will incentivize additional labs to store their data in this format and even to directly acquire data in NWB files to make data immediately shareable (this is already possible for electrophysiological recordings using the Open Ephys GUI; *Siegle et al., 2017*). We envision a future in which it will require less effort for neurophysiologists to comply with community-wide standards than to use their own idiosyncratic conventions because standardized formats serve as a gateway to a host of pre-existing, carefully validated analysis packages.

Standardized metadata conventions are also critical for promoting data reuse. Our surveys are accompanied by extensive white papers, code repositories, and tutorials that detail the minutiae of our methods and tools, beyond the standard 'Methods' section in publications (see *Box 1* for links). For the community at large, a more scalable solution is needed. Standardized and machine-readable metadata needs to extend beyond administrative metadata (describing authors, institutions, and licenses) to include thorough and detailed experimental conditions and parameters in a self-contained manner. As data sharing becomes more widespread, standardization of metadata will be particularly important for reducing 'long tail' effects in which a small number of datasets are reused extensively, while others are disregarded, as observed in the reuse of CRCNS data. To avoid a situation in which publicly available datasets from more focused studies are overlooked, all these studies should be indexed by a single database that can be filtered by relevance, making it much easier for researchers to identify data that is appropriate for their needs. The recently launched Distribute Archives for Neurophysiology Data (DANDI) addresses this concern by enforcing the use of NWB for all shared datasets (*Rübel et al., 2022*). In the two years since the first dataset was uploaded, the archive now hosts more than 100 NWB-formatted datasets accessible via download links, a command-line interface, or within a cloud-based JupyterHub environment.

The human neuroimaging field has faced similar challenges. To address the lack of standardization across public datasets, the community spearheaded the development of the Brain Imaging Data Structure (BIDS), a set of schemas for storing MRI volumes along with associated behavioral and physiological measures (*Gorgolewski et al., 2016*). NWB shares many features of the BIDS standard, including a hierarchical structure, separation of raw and derived data, and support for extensions. BIDS was essential for the success of *OpenNeuro*, a public neuroimaging data archive which, as of 2021, included data from over 20,000 subjects (*Markiewicz et al., 2021*). Given the related aims of OpenNeuro and DANDI, there are many opportunities for the leaders and maintainers of these resources to learn from one another.

While the adoption of consistent data formatting conventions is a welcome development, there are also benefits to greater standardization of protocols, hardware, and software used for data collection. One way this can be achieved is through coordinated cross-laboratories experiments, such as those implemented by the International Brain Laboratory (IBL), a consortium that uses Neuropixels to survey responses across the entire mouse brain in a visual decision-making task (*Abbott et al., 2017*; *Ashwood et al., 2022*). It can also be beneficial to carry out smaller-scale studies on infrastructure built for surveys, as we have done as part of the "OpenScope" project. OpenScope allows members of the wider community to propose experiments to be run by Allen Brain Observatory staff (*Gillon et al., 2023*; *Mayner et al., 2022*; *Prince et al., 2021*). This lowers the barriers to generating high-quality, standards-compliant data, especially for labs whose work is primarily computational. Similarly,

the IBL is now entering a phase in which member laboratories conduct more focused studies that take advantage of existing rigs and data pipelines.

To encourage data sharing, the field of neurophysiology also needs greater standardization in the way data mining is tracked and credited. Digital object identifiers (DOIs) are an essential first step; we regret not making them an integral part of the Visual Coding data releases. However, they have not solved the problem of discovering reuse as they are not always included in publications. It is more common to include a reference to the original paper in which the dataset was described, but this makes it difficult to distinguish instances of reuse from other types of citations. Currently the onus is on those releasing the data to keep track of who accesses it. To take one example, the **cai-1** calcium indicator calibration dataset from the Svoboda Lab at HHMI Janelia Research Campus (*GENIE Project, 2015*) only has five citations tracked in Google Scholar. Yet a deeper dive into the literature reveals that this dataset has been reused in a wide range of publications and conference papers that benchmark methods for inferring spike rate from calcium fluorescence signals, of which there are likely over 100 in total. Many of these papers only cite the original publication associated with this dataset (*Chen et al., 2013*), refer to the repository from which the data was downloaded (CRCNS), or do not cite the data source at all. The lack of an agreed-upon method for citing datasets (like we have for journal articles) is a loss for the community as it hinders our ability to give appropriate credit to those responsible for collecting widely used datasets. A simple, widely accepted method for citing data would benefit all authors as it has been shown that publications within the astrophysics community that provide links to the underlying data gain more citations on average than those that do not (*Dorch et al., 2015*).

## Cloud-based analysis tools

To enable more efficient data mining, end users should ideally not need to download data at all. This is particularly true as the volume of data keeps on growing (e.g., a single Allen Brain Observatory Neuropixels session generates about 1.2 TB of raw data). Therefore, the goal should be to bring users to the data, rather than the data to users. This is supported by our interviews with end users who cited slow download speeds as a key challenge.

Generic analysis tools, such as Amazon's SageMaker and Google's Colab, already make it possible to set up a familiar coding environment in the cloud. However, we are most excited about tools that lower the barriers and the costs of cloud analysis for scientists. Some of the most promising tools include DataJoint (*Yatsenko et al., 2015*), DandiHub, NeuroCAAS (*Abe et al., 2022*), Binder, and Code Ocean (many of which are built on top of the powerful Jupyter platform). All of these are aimed at improving the reproducibility of scientific analyses, while shielding users from the details of configuring cloud services.

Cloud-based analysis is not a panacea. Although individual tools can be vendor-agnostic, there will be a push to centralize around a single cloud platform, given the high cost transferring data out of cloud storage. This could lead to a single company monopolizing the storage of neurophysiology data; it would therefore be prudent to invest in a parallel distribution system that is controlled by scientists (*Saunders and Davis, 2022*). In addition, it is (perhaps not surprisingly) notoriously easy for unwary users to provision expensive cloud computing resources; a single long-running analysis on a powerful cloud workstation could exhaust a lab's entire annual budget without safeguards in place. Despite these drawbacks, we believe that a move to cloud-based analysis will be essential for reducing the friction involved in adopting new datasets. We plan to move toward supporting a cloud-native sharing model more directly in our upcoming data releases.

## Fostering a culture of data reuse

The value of open data is best realized when it is conveniently accessible. Whether this involves new discoveries or comparing results across studies, data mining is vital for progress in neuroscience, especially as the field as a whole shifts toward more centralized 'Observatories' for mice and non-human primates (*Koch et al., 2022*). The BRAIN Initiative has invested considerable resources in advancing instruments and methods for recording a large number of neurons in more sophisticated behavioral contexts. Yet the analytical methods for understanding and interpreting large datasets are lagging, as many of our theoretical paradigms emerged from an era of small-scale recordings (*Urai et al., 2022*).

In order to develop theories that can explain brain-wide cellular neurophysiology data, it is critical to maximize data reuse.

This poses a set of challenges. Any time a scientist uses a new dataset, they must comprehend both how to access and manipulate it *and* decide whether it is appropriate for their question. The latter is the actual scientific challenge, and is where scientists should expend the bulk of their energy. To facilitate this, we first need increased compliance with standards and enhanced tooling around those standards. The more straightforward and intuitive it is to analyze a particular dataset, the more likely it is to be reused. The full burden of refining and adhering to these standards should not fall on the good intentions of individual researchers; instead, we need funding agencies and institutions to recognize the value of open data and allocate resources to facilitate the use of such standards. Everyone benefits when scientists can focus on actual biology rather than on the technical challenges of sharing and accessing data. Second, we need our evaluation of data reuse to ensure that researchers have identified data assets pertinent to their questions and have accounted for the limitations of an experimental paradigm. For instance, we have shown that a naïve comparison of cellular properties measured in the same visual areas across our Neuropixels and two-photon calcium imaging datasets reveals substantial discrepancies (*Siegle et al., 2021b*). These can only be reconciled by accounting for the bias inherent in each recording modality, as well as the data processing steps leading to the calculation of functional metrics. Effective data reuse requires that we, as a field, focus more of our energies on better communicating these important technical factors and holding researchers accountable for understanding them when they analyze someone else's data.

Neuroscientists have traditionally been taught to address questions by collecting new data. As data sharing becomes more prevalent, neuroscientists' first instinct should instead be to search for existing data that may offer insights into the problem at hand, whether or not it was originally intended for this purpose. Even in situations where the 'perfect' dataset does not yet exist, it is likely that researchers can exploit available data to refine a broad question into one that is more focused, and thus experimentally more tractable. Just as young scientists are trained to discover, interpret, and cite relevant publications, it is imperative that they are also taught to effectively identify, evaluate, and mine open datasets.

## Acknowledgements

We thank all the members of Transgenic Colony Management, Lab Animal Services, Neurosurgery & Behavior, Imaging and Neuropixels Operations Teams, Materials & Process Engineering, Information Technology, and Program Management that cared for and trained the animals, built and staffed the instruments, processed the brains, and wrangled the data streams. We thank Allan Jones for providing an environment that nurtured our efforts and the Allen Institute founder, Paul G Allen, for his vision, encouragement, and support. This research was funded by the Allen Institute. We thank Amazon Web Services for providing free cloud data storage as part of the Open Data Registry program. We thank Hilton Lewis for his insights into data sharing in the astronomy community and Fritz Sommer for providing information about CRCNS. We thank Bénédicte Rossi for illustrating the different use cases for our data. We thank Huijeong Jeong, Chinmay Purandare, Noam Nitzan, Carsen Stringer, Shabab Bakhtiari, Aman Saleem, Marius Schneider, and Jorrit Montijn for providing feedback about their user experiences. We thank Karel Svoboda, David Feng, Jerome Lecoq, Shawn Olsen, Stefan Mihalas, Anton Arkhipov, and Michael Buice for feedback on the manuscript.

## Additional information

### Funding

| Funder | Grant reference number | Author |
|---|---|---|
| Allen Institute | | Saskia EJ de Vries<br>Joshua H Siegle<br>Christof Koch |

| Funder | Grant reference number | Author |
|--------|------------------------|--------|

The funders had no role in study design, data collection and interpretation, or the decision to submit the work for publication.

## Author contributions
Saskia EJ de Vries, Joshua H Siegle, Christof Koch, Conceptualization, Writing – original draft, Writing – review and editing

## Author ORCIDs
Saskia EJ de Vries ⓘ http://orcid.org/0000-0002-3704-3499
Joshua H Siegle ⓘ http://orcid.org/0000-0002-7736-4844

## Decision letter and Author response
Decision letter https://doi.org/10.7554/eLife.85550.sa1
Author response https://doi.org/10.7554/eLife.85550.sa2

---

## Additional files

### Supplementary files
- MDAR checklist
- Source data 1. List of publications using Allen Brain Observatory Visual Coding data as of 2022.

### Data availability
The Allen Brain Observatory Visual Coding datasets are available at https://portal.brain-map.org/explore/circuits. The list of papers reusing these datasets is provided in *Source data 1*.

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
