## [Editor Report]

This article presents an important review of data-sharing efforts in neurophysiology, with a focus on data released by the Allen Institute for Brain Science. The article offers perspectives from the users of such shared data, and makes a compelling case that data sharing has already advanced research in neuroscience. There are valuable insights here for producers and users of neurophysiology data, as well as the funders that support all those efforts.

---

## [Decision Letter]

**Decision letter after peer review:**

Thank you for submitting your article "Sharing Neurophysiology Data from the Allen Brain Observatory: Lessons Learned" for consideration by *eLife*. Your article has been reviewed by 2 peer reviewers, including Markus Meister as the Reviewing Editor and Reviewer #1, and the evaluation has been overseen by Joshua Gold as the Senior Editor. The following individual involved in the review of your submission has agreed to reveal their identity: Jerry L Chen (Reviewer #2).

Essential revisions:

*Reviewer #1 (Recommendations for the authors):*

1. First sentence, "Making all data for any observation or experiment openly available is a defining feature of empirical science (e.g., nullius in verba, the motto of the Royal Society)." This doesn't seem quite right. Empirical science has been around for several centuries, and the full sharing of all data has not been a defining feature or an expectation during almost all that time. As the authors concede later on, it is not even commonly practiced today. Does that mean those practitioners are not engaged in empirical science? The motto of the Royal Society is about valuing evidence from experiments over the dogma from authority figures, not about publishing the raw data from experiments.

Recommendation: Please revise as needed.

2. p.1, last par, comparison to other data sharing efforts "facilitating numerous advances".

– In this comparison, CRCNS is unlike all the other instances; in fact, it is the only one that can be said to facilitate numerous advances. See here for a citation count (up to 2019): https://crcns.org/publications: Over 100 journal articles and 25 preprints, various textbook chapters, summer courses, etc. This matches or exceeds the impact of AI data sharing (of course over a longer time period), so it should serve as a reference point.

– By comparison, for example, the Buzsaki databank Petersen 2020 has only one documented instance of reuse outside the Buzsaki lab (https://www.biorxiv.org/content/10.1101/2022.04.01.486754v1). Of course, Buzsaki was also a major contributor to the CRCNS site.

– Given the success of data sharing by CRCNS, it could serve as a reference for some of the conclusions in the rest of the article. For example, CRCNS did not insist on a standard data format, yet apparently, that was not a major obstacle to reuse.

Recommendation: Please expand a comparative evaluation of contributions to data-sharing from CRCNS.

3. p.2, "learning how to analyze open datasets can take substantial effort", and p.12: "Scientists cannot afford to spend time trying to understand obscure data formats or deciphering which experimental conditions were used" and "We must be able to focus on actual biology rather than on the technical challenges of sharing and accessing data"

– Scientific research in general takes substantial effort and learning. Why should that not be true for this particular research activity, namely re-purposing the results of other people's experiments? These sections give the impression that data-sharing should be cost-free for the recipient and that all the effort is to be borne by the provider.

– Is it unreasonable for computational neuroscientists to spend effort on this? After all, experimenters invest considerable time trying to understand Nature's "obscure data formats", and building instruments to read those formats. Similarly, data miners should expect to spend some effort mining, given that the data were not acquired with their needs in mind.

– Has this issue really been an obstacle to progress? Again, see the CRCNS experience.

Recommendation: Consider how the effort involved in data sharing should be distributed between the provider and the receiver of the data.

4. p.10, "For the community at large, a more scalable solution is needed": This section acknowledges that small laboratories don't have the same resources as large organizations (AI, IBL). But the recommendation here seems to be that small players just need to man up and try harder: get your metadata organized according to standards, improve your quality control, use prescribed data formats so you can join DANDI, etc. Yet, experience suggests that all this may not be essential. See the success of CRCNS, which was fed entirely by small labs, and allowed heterogeneous data formats and presentations of metadata.

Recommendation: The article could be more informative if you compared the reuse of AI data to that of CRCNS data.

5. p.12, "identify and prevent inappropriate reuse": Did you discover such instances in your survey of the 100 reuse cases?

Recommendation: If so, it would be instructive to report their nature, even without identifying the papers by name.

*Reviewer #2 (Recommendations for the authors):*

I would encourage the authors to take a little time to survey and include the perspectives of the authors in the community that re-used the data set.

---

## [Author Response]

Essential revisions:Reviewer #1 (Recommendations for the authors):1. First sentence, "Making all data for any observation or experiment openly available is a defining feature of empirical science (e.g., nullius in verba, the motto of the Royal Society)." This doesn't seem quite right. Empirical science has been around for several centuries, and the full sharing of all data has not been a defining feature or an expectation during almost all that time. As the authors concede later on, it is not even commonly practiced today. Does that mean those practitioners are not engaged in empirical science? The motto of the Royal Society is about valuing evidence from experiments over the dogma from authority figures, not about publishing the raw data from experiments.Recommendation: Please revise as needed.

We have revised the beginning of the abstract as follows: “Nullius in verba (‘trust no one’), chosen as the motto of the Royal Society in 1660, implies that independently verifiable observations—rather than authoritative claims—are a defining feature of empirical science. As the complexity of modern scientific instrumentation has made most exact replications cost prohibitive, sharing data is now essential for ensuring the trustworthiness of one’s findings.”

2. p.1, last par, comparison to other data sharing efforts "facilitating numerous advances".– In this comparison, CRCNS is unlike all the other instances; in fact, it is the only one that can be said to facilitate numerous advances. See here for a citation count (up to 2019): https://crcns.org/publications: Over 100 journal articles and 25 preprints, various textbook chapters, summer courses, etc. This matches or exceeds the impact of AI data sharing (of course over a longer time period), so it should serve as a reference point.– By comparison, for example, the Buzsaki databank Petersen 2020 has only one documented instance of reuse outside the Buzsaki lab (https://www.biorxiv.org/content/10.1101/2022.04.01.486754v1). Of course, Buzsaki was also a major contributor to the CRCNS site.– Given the success of data sharing by CRCNS, it could serve as a reference for some of the conclusions in the rest of the article. For example, CRCNS did not insist on a standard data format, yet apparently, that was not a major obstacle to reuse.Recommendation: Please expand a comparative evaluation of contributions to data-sharing from CRCNS.

We have expanded our coverage of CRCNS in the introduction (p. 2). In addition to describing the contents of this database in more detail, we examined all of the publications listed on the CRCNS website to determine the distribution of reuse across datasets. We found that the most reuse is skewed toward a small number of datasets, and fewer than 20% of datasets have been reused at all.

3. p.2, "learning how to analyze open datasets can take substantial effort", and p.12: "Scientists cannot afford to spend time trying to understand obscure data formats or deciphering which experimental conditions were used" and "We must be able to focus on actual biology rather than on the technical challenges of sharing and accessing data"– Scientific research in general takes substantial effort and learning. Why should that not be true for this particular research activity, namely re-purposing the results of other people's experiments? These sections give the impression that data-sharing should be cost-free for the recipient and that all the effort is to be borne by the provider.– Is it unreasonable for computational neuroscientists to spend effort on this? After all, experimenters invest considerable time trying to understand Nature's "obscure data formats", and building instruments to read those formats. Similarly, data miners should expect to spend some effort mining, given that the data were not acquired with their needs in mind.– Has this issue really been an obstacle to progress? Again, see the CRCNS experience.Recommendation: Consider how the effort involved in data sharing should be distributed between the provider and the receiver of the data.

We believe that the proliferation of data sharing standards will benefit data producers as much as data consumers. As the tools for data packaging mature, less effort will be required to format and distribute one’s data. And the easier it is to read and access new datasets, the greater the impact they are likely to have.

To make these points clear, we have added the following text to the discussion: “Any time a scientist uses a new dataset, they must comprehend both how to access and manipulate it *and* decide whether it is appropriate for their question. The latter is the actual scientific challenge, and is where scientists should expend the bulk of their energy. To facilitate this, we first need increased compliance with standards and enhanced tooling around those standards. The more straightforward and intuitive it is to analyze a particular dataset, the more likely it is to be reused. The full burden of refining and adhering to these standards should not fall on the good intentions of individual researchers; instead, we need funding agencies and institutions to recognize the value of open data and allocate resources to facilitate the use of such standards. Everyone benefits when scientists can focus on actual biology rather than on the technical challenges of sharing and accessing data.”

4. p.10, "For the community at large, a more scalable solution is needed": This section acknowledges that small laboratories don't have the same resources as large organizations (AI, IBL). But the recommendation here seems to be that small players just need to man up and try harder: get your metadata organized according to standards, improve your quality control, use prescribed data formats so you can join DANDI, etc. Yet, experience suggests that all this may not be essential. See the success of CRCNS, which was fed entirely by small labs, and allowed heterogeneous data formats and presentations of metadata.Recommendation: The article could be more informative if you compared the reuse of AI data to that of CRCNS data.

We now include statistics about the reuse of CRCNS data in the introduction. We found that 122 out of 150 datasets hosted by CRCNS have no documented instances of reuse. While we don’t know the reasons for this, the lack of data and metadata standards are likely to play a role. Our recommendation goes beyond “man up” and specifically calls for enhanced funding and tooling to facilitate such efforts.

5. p.12, "identify and prevent inappropriate reuse": Did you discover such instances in your survey of the 100 reuse cases?Recommendation: If so, it would be instructive to report their nature, even without identifying the papers by name.

We have not yet found any examples of this for the Allen Brain Observatory, but the potential for inappropriate reuse was cited by some of our colleagues as a major reason why they are hesitant to share their own data.

Reviewer #2 (Recommendations for the authors):I would encourage the authors to take a little time to survey and include the perspectives of the authors in the community that re-used the data set.

See the new “User experience” section, which incorporates feedback from members of the community.